# Lifestage Sex-Specific Genetic Effects on Metabolic Disorders in an Adult Population in Korea: The Korean Genome and Epidemiology Study

**DOI:** 10.3390/ijms231911889

**Published:** 2022-10-06

**Authors:** Young-Sang Kim, Yon Chul Park, Ja-Eun Choi, Jae-Min Park, Kunhee Han, Kwangyoon Kim, Bom-Taeck Kim, Kyung-Won Hong

**Affiliations:** 1Department of Family Medicine, CHA Bundang Medical Center, CHA University, Seongnam-si 13496, Korea; 2Department of Family Medicine, Wonju Severance Christian Hospital, Yonsei University Wonju College of Medicine, Wonju-si 26426, Korea; 3Department of Medical Education, Yonsei University Wonju College of Medicine, Wonju si 26462, Korea; 4Department of Healthcare Technology, Theragen Bio Co., Ltd., Pangyoyeok-ro 240, Seongnam-si 13493, Korea; 5Department of Family Medicine, Gangnam Severance Hospital, Yonsei University College of Medicine, Seoul 06273, Korea; 6Department of Family Medicine, Seonam Hospital, Seoul 08049, Korea; 7Department of Family Practice and Community Health, Ajou University Hospital, Suwon-si 16499, Korea; 8Department of Family Medicine, Myungju Hospital, Yongin 17050, Korea

**Keywords:** premenopause, perimenopause, postmenopause, metabolic disorders, genome-wide association studies

## Abstract

Although many genome-wide association studies (GWASs) have evaluated the association with metabolic disorders, the current study is the first attempt to analyze the genetic risk factors for various metabolic disorders according to sex and age groups of the life course in Korean adults. A total population of 50,808 people were included in this GWAS. The genetic traits for eight metabolic phenotypes were investigated in peri-, and postmenopausal women compared to a younger group or men of corresponding age groups. The metabolic phenotypes include general obesity, abdominal obesity, hypertension, type 2 diabetes, hypercholesterolemia, hypertriglyceridemia, hypo-high-density lipoprotein cholesterolemia, and metabolic syndrome. In the total participants, GWAS results for eight metabolic phenotypes found 101 significant loci. Of these, 15 loci were the first reported to be associated with the risk of metabolic disorder. Interestingly, some of the significant loci presented the association with the various phenotypes, which presented when there was a correlation between phenotypes. In addition, we analyzed divided by gender and age (young adult, peri-menopausal group, older adult), and specifically identified specific loci in peri-menopausal women. Meanwhile, several genetic factors associated with metabolic disorders were newly reported in our study. In particular, several genes were significantly associated with one of the metabolic phenotypes in only a single specific group. These findings suggest that menopausal transition rather than aging itself potentiates the influence of genetic risks on metabolic disorders. In addition, some genetic loci with low frequencies may play a role in the metabolic disturbances in a specific sex and age group. The genetic traits derived from our study may contribute to understanding the genetic risk factors for metabolic disorders in the Korean population.

## 1. Introduction

Unlike men, during the aging process women experience marked hormonal changes associated with menopause. During this transition, women experience specific symptoms like hot flashes, mood irritability, and vaginal dryness; they are also exposed to an increased chance of cardiovascular disease (CVD). After the menopausal transition, the CVD risk significantly increases to the rate seen in men [1,2]. CVD is a leading cause of morbidity and mortality worldwide [3]. Hence, it is important for perimenopausal women to recognize risk factors and address them to prevent CVD and reduce cardiovascular complications.

Established risk factors for CVD include hypertension, type 2 diabetes mellitus (T2D), dyslipidemia, and smoking; they also include uncontrollable factors like age, sex, and race [4]. Metabolic syndrome (MetS) is generally defined as a clustering of metabolic disturbances such as abdominal obesity, dyslipidemia, hypertension, and the impaired regulation of glucose metabolism [5]. MetS confers a two-fold increase in the risk of CVD development during the subsequent 5 to 10 years of a patient’s life [6]. CVD and its related metabolic disorders are strongly affected by aging. The prevalence of CVD and related disorders, including hypertension, coronary heart disease, heart failure, and stroke, increases from 40% at 40–59 years of age to 79–86% among those 80 years or older [7]. MetS prevalence also varies considerably between younger and older adult populations [8,9,10]. One survey conducted in Korea found that prevalence varies according to age groups and sex [11]. MetS prevalence increases sharply after the sixth decade of life and continues to increase, especially in women. This finding suggests that menopause strongly affects the development of metabolic derangements.

Techniques used for genotyping have changed markedly over the last 10 years; genotyping large sets of genetic variants in individuals can now be performed at a lower cost [12,13]. Genetic factors that contribute to CVD and its risk factors have been identified using genome-wide association studies (GWASs) [14]. A GWAS implicated specific loci as risk variants for coronary artery disease in an East Asian population [15]. GWASs have also found associations between cardiovascular risk factors and MetS in Korean populations [16,17,18].

Although genes associated with metabolic disorders can differ according to specific lifetime periods, most GWASs for CVD and its risk factors have regarded age and sex as potential confounding variables that are adjusted during the analysis. Considering that age at menopause is strongly affected by genetic factors in women [19,20], and that women experience rapid increases in metabolic abnormalities during the perimenopausal period [21], the genetic traits of metabolic disorders may be different before and after the menopausal transition. First, we conducted a GWAS for metabolic disorders including (1) general obesity, (2) abdominal obesity, (3) hypertension, (4) T2D, (5) hypercholesterolemia, (6) hypertriglyceridemia, (7) hypo-high-density lipoprotein (HDL) cholesterolemia, and (8) MetS using a large dataset obtained from the Korean Genome and Epidemiology Study [22]. We also investigated genetic traits in peri- and postmenopausal women compared to younger groups of women and to men in corresponding age groups.

## 2. Results

### 2.1. Clinical Characteristics of Study Groups

Subjects were categorized into six groups according to three age groups and both sexes (Figure 1). The total population consisted of 19,595 (38.6%) males and 31,213 (61.4%) females. The numbers of women undergoing menopausal transition and of men who corresponded in age were 14,453 and 7179, respectively (Table 1). The proportions for general obesity, abdominal obesity, hypertension, T2D, hypercholesterolemia, hypertriglyceridemia, hypo-HDL cholesterolemia, and MetS were 32.3%, 16.0%, 26.4%, 8.6%, 11.2%, 12.7%, 28.2%, and 20.8%, respectively.

The population of females had lower proportions of current smokers and alcohol consumers. Except for total cholesterol and HDL cholesterol, the proportions of most of the anthropometric and biochemical traits were lower in the female group than in the male group. Similarly, most of the metabolic disorder phenotypes were more frequent in the male than in the female group, except hypercholesterolemia and hypo-HDL cholesterolemia. Between the sexes in each age subgroup (young adult group, perimenopausal group, and older age group), the trends were similar to the whole subjects group. In both sex groups, the proportions of smokers and alcohol consumers gradually decreased from the youngest age group to the oldest age group, but routine exercise was increased in the older age group. The findings for anthropometric measurements, biochemical assays, and metabolic disorder frequencies revealed more prominent differences in the female groups from the young adult group to the older age group than the male groups. For example, the general obesity proportions were slightly decreased with aging in men, but were greatly increased in women.

### 2.2. Overall Results of the GWASs

We used genome-wide significant *p*-value criteria (*p* < 5 × 10^−8^) or the bonferroni correction *p*-value criteria (*p* < 6.27 × 10^−9^), and selected the top significant single nucleotide polymorphism (SNP) among the clustered SNPs (Table 1 and Figure 2). The results for significantly clustered loci were presented using the alphabet ‘L’ with corresponding serial numbers (Appendix A). Each figure consists of subgroup results using a Manhattan plot for the entire subject group and Miami plots for male and female groups, young adult male and female groups, perimenopausal male and female groups, and older adult male and female groups. The results for the top significant SNPs in each significant cluster region are presented in Appendix A. A total of 101 significant loci were detected in this GWAS study: eight for general obesity (GO); four for abdominal obesity (AB); 19 for hypertension (HTN); nine for T2D (DM); 19 for hypercholesterolemia (CH); 14 for hypertriglyceridemia (TG); 20 for hypo-HDL cholesterolemia (HDL); and eight for MetS (Appendix A). 

### 2.3. Study Group-Specific Associations

The results for group-specific association genes are presented in Figure 3, Table 2, Table 3, and Appendix A. The most significant association with general obesity was found in the GO-L1:*SEC16B* gene region (*p*-value = 1.21 × 10^−15^); there were more significant effects among the female group comparisons than among the male group comparisons (Figure 2 and Appendix A). The trends were greatly increased from the peri-menopausal female group (*p*-value = 4.03 × 10^−8^) to the older adult group (*p*-value = 1.35 × 10^−6^). The *SEC16B* gene region was also the top significant gene region associated with abdominal obesity (*p*-value = 3.20 × 10^−9^), and the trends for the sex and age groups were similar to those for general obesity (Figure 2 and Appendix A). The most significant association with abdominal obesity was found for the AB-L3:*ONECUT1* gene of the peri-menopausal male group (*p*-value = 8.64 × 10^−10^; Table 2 and Appendix A). The other interesting association with abdominal obesity was found in the AB-L4:*XKR3* gene region. The SNP (rs187426985) of the AB-L4:*XKR3* gene is a low- frequency SNP present in East Asian populations. This association was found in the perimenopause-corresponding male group, the young adult female group, and the older adult female group. The odds ratios (1.6–1.7) were relatively higher than those for the conventional common disease GWAS results (Table 4).

This study was the first to find 15 of the significant loci associated with metabolic disorders: GO-L3:*ITIH4* for general obesity; AB-L4:*XKR3* for abdominal obesity; HTN-L6:*HLAB* and HTN-L18:*LRRC30* for HTN; DM-L1:*3p13* and DM-L9:*7p12* for T2D; TG-L3:*HSD17B13*, TG-L4:*VARS*, and TG-L11:*FSD2* for hypertriglyceridemia; HDL-L3:*TSBP1*, HDL-L7:*TRPS1*, and HDL-L8:*9q21.31* for hypo-HDL cholesterolemia; MetS-L1:*6q21*, MetS-L2:*COA1*, and MetS-L3:*HTR5A* for MetS (Table 4).

Among the hypertension results, the most significant association was found in the HTN-L5:*FGF5* gene region, and the trends were consistent in all study groups (Table 2; Appendix A). This result suggested that the *FGF* gene was the essential marker for Korean hypertension estimation without sex and age effects. In contrast, the associations for the HTN-L4:*CACNA1D*, HTN-L6:*HLAB*, HTN-L8:*CYP11B1*, and HTN-L11:*LSP1* gene regions were female-specific. The association trends were found in the perimenopausal female group and the older female group. These results suggested that the *CACNA1D* gene, *HLAB* gene, and *CYP11B1* markers should be examined for women in post-menopause to effectively care for their cardiovascular health. The HTN-L15:*NEDD4* gene region was significant only in the older adult female group. In contrast, the HTN-L14:*ALDH2* gene region had a male-specific association. Another association was found in the HTN-L17:*LRRC30* gene region for the young female group only (Figure 2 and Appendix A). The SNP (rs74981150) of the HTN-L17:*LRRC30* gene was the East Asian-specific variant and greatly increased hypertension risk (odds ratio = 2.987, *p*-value = 1.49 × 10^−8^; Table 2), possibly because of the relationship with gestational hypertension. The results for the HTN-L18:*RGL3* gene region were different between male and female groups. Among the male groups, the young adult male group and the perimenopausal male group showed significant associations with hypertension, but the older adult male group did not. Among the female groups, the perimenopausal female group and the older adult group showed significant associations with hypertension; the association with the young adult female group was not significant.

Among the T2D associations, the most significant locus was DM-L2:*CDKAL1*, and the locus consistently increased the T2D risk in both sexes and in all age groups (Table 2 and Appendix A). We identified two loci (DM-L1:*3p13* and DM-L9:*17p12*) associated with the young adult female group. The other loci are well-known findings from Korean GWASs and GWASs for other ethnic groups. Among them, two loci (DM-L7:*KCNQ1* and DM-L8:*HECTD4*) were significant for the postmenopausal groups to the older adult groups. The DM-L3:*PAX4* locus is an Asian-specific polymorphism and increases the T2D risk by a factor of 1.4.

We also performed three GWAS analyses for hyper-cholesterolemia, triglyceridemia, and hypo-HDL-cholesterolemia. The *APOA5* locus was associated with three lipid disorders, and the associations were greatly and very significantly increased for all of them (hypercholesterolemia = OR: 1.11, *p*-value = 1.37 × 10^−8^; triglyceridemia = OR: 1.89, *p*-value = 5.01 × 10^−228^; and hypo-HDL-cholesterolemia = OR: 1.83, *p*-value = 3.80 × 10^−164^; Table 2). The CH-L1:*PCSK9* and CH-L3:*APOB* gene polymorphisms were Korean and East Asian- specific and had significant effects on all study groups (Table 2 and Appendix A).

The CH-L19:*TOMM40* gene polymorphism had the most significant association with hypercholesterolemia risk in all study groups (Table 2 and Appendix A). The *GCKR* loci were significant for hypercholesterolemia and hypertriglyceridemia (CH-L4 and TG-L2, respectively); the *APOB* and *ABCA1* loci were significantly associated with both hypercholesterolemia (CH-L3:*APOB* and CH-L10:*ABCA1*, respectively) and hypo-HDL cholesterolemia (HDL-L2:*APOB* and HDL-L9:*ABCA1*, respectively).

CH-L8:*UBXN2B-CYP7A1* was associated with significant risk for hypercholesterolemia in the young adult male group. Significant associations for the TG- L10:*ALDH2* polymorphism (i.e., the alcohol flushing gene) were found only for the male groups (Table 2 and Appendix A). Significant risks for hypo-HDL-cholesterolemia for the perimenopausal female group and older adult female group were revealed for HDL- L5:*KLF14* and HDL-L10:*JMJDIC* (Table 2 and Appendix A). There was a significant association between HDL-L8:*9q21.31* and hypo-HDL-cholesterolemia in the perimenopausal male group. The HDL-L16:*CETP* gene had the most significant association with hypo-HDL-cholesterolemia for all study groups (Table 2 and Appendix A).

Among the loci for MetS, genes like *LPL*, *APOA5*, *ALDH2*, *CETP*, and *APOC1* overlapped for the different types of phenotypes (Table 2 and Appendix A). *FRMD4B* and *COA1* were associated with MetS in men from older and younger age groups, respectively.

## 3. Discussion

We examined genetic associations between sex and age groups. In general, men and women had differences in the distribution of lipoprotein lipids and some enzymes that control lipoprotein metabolism [23]. There were also between-sex differences in metabolic phenotypes in this study population. The changes in prevalence of various metabolic phenotypes were more prominent in women than in men. These results suggested that some genetic factors were affected by sex and aging. We assumed that the marked hormonal change during menopausal transition likely affected the activities of transcription factors and changed genetic traits. The group-specific GWAS analyses detected 101 significant gene loci, including 15 that were newly identified.

### 3.1. Significant Associations in Peri- and Postmenopausal Women

We found specific gene regions with significant associations in peri- and postmenopausal women. The genetic polymorphisms of the *SEC16B* gene region showed significant associations with obesity; *SEC16B* is a well-known obesity-related genetic variant [24]. *SEC16B* is also a gene region associated with age at menarche and other reproductive traits [25,26]. Thus, given that *SEC16B* is related to female reproductive function, the association between this locus and general obesity later life in women may be explainable.

The genetic polymorphisms of the *CACNA1D*, *HLAB*, *CYP11B1*, and *LSP1* gene regions showed significant association with hypertension [27,28,29,30,31]. In particular, *CYP11B1* affects aldosterone and cortisol synthesis and altered regulation of the pituitary-adrenal axis at the terminal steps, which causes changes in ACTH levels and may increase the risk of hypertension [28]. Variants of the *HLA-B* gene that are associated with the risk of hypertension based on hyper-inflammatory status have been found and are related to disease-causing pulmonary hypertension caused by thrombosis [32]. The genetic polymorphisms of the *JMJD1C* and *KLF14* gene regions showed significant associations with HDL cholesterolemia. The *JMJD1C* gene is involved in sex steroid hormone metabolism and can affect thyroid hormone metabolism [33]. The *KLF14* gene is a major factor in placenta formation and is involved in changes in gene expression patterns that control placental growth [34]. Several genetic factors associated with metabolic disorders were newly found in our study. The SNP, rs187426985 on *HSFY1P1* was related to abdominal obesity. This gene is a non-coding pseudogene and is associated with Cat Eye syndrome [35]. The *XKR3* gene is nearby to *HSFY1P1*, which is predominantly expressed in the testes [36]. Associations between this locus and metabolic disorders have not been reported*. MIR6891* is an RNA gene and is affiliated with the miRNA class. Previously, this gene was known to be associated with diffuse large B cell lymphoma [37]. Expression levels of some microRNAs, including miR-6891-5p, may have vital roles in the physiological and pathological progression of coronary artery disease [38]. Our study was the first to find a low prevalence of hypertension in women with variants of this gene.

### 3.2. Newly Identified Metabolic Syndrome Gene Loci

This study was the first to find several genetic factors associated with types of dyslipidemia. Associations between hypertriglyceridemia and the SNPs on *HSD17B11*, *VARS*, and *FSD2* were not previously reported. Although a few SNPs associated with dyslipidemia have been found, the genes with those SNPs have not previously been reported. Regarding associations with hypertriglyceridemia, SNPs on the genes of *BCL7B*, *FADS2*, and *TM6SF2* were found in previous studies [39,40,41,42,43]. *HSD17B11* codes a protein, hydroxysteroid 17-beta dehydrogenase 11. This gene may participate in the androgen metabolism during steroidogenesis. The metabolism may affect sex differences relevant to the role of this gene. In our study, the association between this SNP and hypertriglyceridemia was significant only in women. The *VARS1* gene codes mitochondrial valyl-TRNA synthetase 1, which catalyzes the attachment of valine to tRNA(Val) for mitochondrial translation. The *FSD2* gene codes fibronectin type III and SPRY domain-containing protein 2. Given that the associations between these genes and triglyceride were not previously reported, the mechanisms that affect their relationships remain unclear. The genes *TSBP1*, *TRPS1*, and *JMJD1C* code their downstream proteins. *TRPS1* regulates DNA-binding transcription factor activity and sequence-specific DNA binding, and *JMJD1C* participates in histone demethylation; our novel findings suggested that metabolic defects in HDL cholesterol are related to DNA binding activity.

Some SNPs were located on genes like *TLE4*, *TFAP2A*, *ONECUT1*, *FRMD4B*, and *COA1*. Associations between these genes and metabolic disorders have not been reported. We found that the SNP rs78000468 on this gene was associated with hypo-HDL cholesterolemia in middle-aged men. The Transducin-like enhancer of split 4 (TLE4) affects the Wnt signaling pathway. Cholesterol has a regulatory role in Wnt signaling [44]. However, the association between TLE4 and cholesterol has not previously been reported. The variant frequency of this SNP is very low in Europe and America; the MAF in our population was only 0.032. The SNP rs569210477 variant on the *TFAP2A* gene was associated with a higher risk of hypercholesterolemia in young men. Transcription factor activating enhancer binding protein 2 alpha (TFAP2A) has a role in ectoderm differentiation. TFAP2 family members mediate the pro-lipid droplet signal induced by Wnt3a, which suggests that the TFAP2 transcription factor functions as a regulator of lipid droplet biogenesis [38]. An association between this gene and cholesterol has not been reported. This SNP variant was also rare in our population (MAF = 0.015), and in Europe and America (MAF = 0.000).

The SNP rs1899752 variant on *ONECUT1* was associated with a lower risk of abdominal obesity in young men. The gene *ONECUT1* is associated with hepatocellular carcinoma. It also codes hepatocyte nuclear factor 6, which directly interacts with *REV*-*ERB-α* [45]. One study found that the *REV*-*ERB-α* polymorphism is associated with obesity in a male population in Spain [46]. However, the mechanism for the association between this gene and obesity remains unclear.

### 3.3. Confirmed the Previous Associations

Several genetic factors were associated with a specific metabolic disorder, regardless of sex or age group. For example, the association between *CDKAL1* and T2D was significant in most age groups of men and women. *CDKAL1* is a T2D susceptibility gene and has a role in mitochondrial function in adipose tissue [47]. Our study results supported this previous finding.

In this study, types of dyslipidemias were significantly associated with previously reported genetic factors in most age groups of men and women. *APOA5* mainly functions to affect plasma triglyceride levels [34]. Our study also found that this gene was associated with hypertriglyceridemia and hypo-HDL cholesterolemia. The genetic factors such as *APOB*, *LPL*, *LIPC*, *CETP*, and *GCKR* were found in many previous studies [48,49,50,51,52]. Our study findings also support the life-long effects of these genes on dyslipidemia.

The relationship between *MC4R* and obesity was significant only in postmenopausal women. *MC4R* has an essential role in the regulation of feeding and energy balance [53]. Sex-specificity was previously found for this locus with regard to human brain structure and eating behavior [54]. Findings using a mouse model indicated that estradiol stimulates proopiomelanocortin neurons [55], which promotes anorexigenic activity. This connection between these neurons and *MC4R* might affect the higher prevalence of general obesity in older women with *MC4R* gene variants. We found that the SNP rs17244834 on *HMGCR* was associated with hypercholesterolemia in peri- and postmenopausal women. Three-hydroxy-3-methylglutaryl-CoA reductase regulates cholesterol synthesis. Although the SNP rs33761740 has no linkage disequilibrium (LD) with ours, polymorphism in this SNP on *HMGCR* was associated with CHD in female subjects only [56].

Results of a previous meta-analysis supported the findings that, compared with younger women, women in postmenopause tend to have unfavorable lipid profiles [57]. However, the mechanisms that affect changes in lipid profiles are not fully understood, and genetic factors are considered candidate risk factors for these changes. We found several genetic factors strongly associated with types of dyslipidemias. The results of subgroup analyses indicated that the *TOMM40* gene was associated with hypercholesterolemia in peri- and postmenopausal women. *ApoC1* and *ApoE* [58] SNPs on *TOMM40* are associated with longevity and Alzheimer’s disease risk [59]. This gene is also associated with lipid profiles [60]. Estrogen increases hepatic lipase levels [61], regulates hepatic LDL receptors [62], and promotes the liver secretion of cholesterol into bile [63]. Thus, the protective effect of estrogen on hypercholesterolemia may be attenuated in variants of related genes. Several genes were significantly associated with one of the metabolic phenotypes in only a single, specific group. Because a few SNPs were located in intergenic areas, the roles of the genes are unclear. In young women, the rs74981150 and rs57014960 SNP variants were associated with three to four times greater risks for hypertension and T2D, respectively, but their minor allele frequencies (MAFs) were low (0.2–0.3). Because the significance occurred in a specific group, it is reasonable to emphasize the risks of these disorders to this group.

### 3.4. Study Limitations

Our study had some limitations. First, menopause was not defined based on biological evidence. Grouping by age might have misclassified subjects into the wrong groups. However, the menopausal transition does not occur during a specific point in life; it occurs over a multi-year period. Approximately 50 years of age is an important transition period in both men and women. Second, the newly reported genetic markers were not replicated in another population. Studies of other cohorts will be necessary to support the findings about these markers and to investigate the mechanisms of the associations.

The subjects of our study are Koreans, and, therefore, the results of this study may be different in different ethnic groups due to different environmental factors. Therefore, it is necessary that the results of this study be confirmed in different ethnic groups.

### 3.5. Conclusions

This study investigated genetic risk factors for metabolic disorders in a large Korean population. We found new associations between several novel genes and metabolic disorders. Subgroup analyses revealed that some previously reported genes were associated with metabolic disorders, mostly in peri- and postmenopausal women. These results suggested that the genetic traits were associated with the protective effects of female sex hormones against the metabolic disorders. We also found that several SNPs were associated with metabolic disorders in specific subgroups.

## 4. Materials and Methods

### 4.1. Study Participants

The study participants were recruited from a Korean Genome and Epidemiology Study (KoGES) cohort. The detailed description and design of the KoGES can be found in the previously published paper by Kim et al. [22]. In this study, we used the health examination cohort that is a part of the KoGES cohort data (KoGES_HEXA) shared by Korea Biobank. The cohort consists of community-dwelling men and women aged 40–79 years recruited from a national health examinee registry.

A total of 58,701 participants with available genome-wide SNP genotype data were included in the KoGES_HEXA. Of these, we excluded participants who had missing values (*n* = 358) or who had a history of any cancer, thyroid disease, or suffered from surgical menopause (*n* = 7535). Data from 50,808 participants were included in the analyses. Written informed consent was obtained from all subjects. This research project was approved by the Institutional Review Board of Theragen Etex (approval numbers: 700062-20190819-GP-006-02). This study complied with the ethical principles of the Declaration of Helsinki.

### 4.2. Measurement of Anthropometric and Laboratory Data and Definition of Lifestyle Factors

Weight and height were measured to the nearest 0.1 kg and 0.1 cm, respectively, with participants wearing lightweight indoor clothing and no shoes. Body mass index (BMI) was calculated as weight (kg) divided by the square of height (m^2^). Waist circumference was measured to the nearest 0.1 cm at the midpoint between the lower border of the rib cage and the iliac crest at the end stage of normal expiration. Systolic (SBP) and diastolic blood pressure (DBP) were each measured twice using a standardized mercury sphygmomanometer (Baumanometer-Standby; W.A. Baum Co., Inc., New York, NY, USA). Blood samples were obtained on the morning after an overnight fast. Fasting plasma glucose, total cholesterol, triglyceride, and HDL cholesterol were measured using an automatic analyzer (ADIVA 1650, Siemens, Tarrytown, NY, USA).

A regular exerciser was defined as a participant who answered “Yes” to the question, “Do you exercise regularly enough to sweat your body?” Participants who had smoked over 100 cigarettes throughout their lifetimes were considered as smokers. An alcohol consumer was defined as a participant who answered “Yes” to the question, “Do you drink alcohol?”.

### 4.3. Study Phenotypes and Covariates

Considering the mean age at menopause of Korean women [59], the women aged 45–55 years old were regarded as those in the perimenopausal period (perimenopausal women). Men in this age group were defined as perimenopause-corresponding men.

Consequently, those younger than 45 years and older than 55 years of age were assigned to the young adult group and the older age group, respectively.

We used eight metabolic disorders as study phonotypes: (1) General obesity was defined as BMI ≥ 25 kg/m^2^; (2) abdominal obesity was defined as waist circumference ≥ 90 cm in men and ≥85 cm in women [60]; (3) hypertension was defined as SBP ≥ 140 mmHg and/or DBP ≥ 90 mmHg and/or taking anti-hypertensive agents; (4) T2D was defined as fasting glucose ≥ 126 mg/dL and/or taking hypoglycemic agents; (5) hypercholesterolemia was defined as total cholesterol ≥ 240 mg/dL 3; (6) hypertriglyceridemia was defined as triglyceride ≥ 200 mg/dL [61]; and (7) hypo-HDL cholesterolemia was defined as HDL cholesterol < 40 mg/dL in men and <50 mg/dL in women. (8) MetS was defined by the presence of three of more of the following characteristics: (a) abdominal obesity, (b) SBP ≥ 130 mmHg and/or DBP ≥ 85 mmHg and/or antihypertensive medication use, (c) hypo-HDL cholesterolemia, (d) triglyceride ≥ 150 mg/dL, (e) elevated fasting glucose ≥ 100 mg/dL and/or taking hypoglycemic agents. We used the general information on age, sex, and lifestyle behaviors as covariates in all analyses.

### 4.4. Study Genotypes

Cohort population genotyping was based on KoGES_HEXA cohort results. The genotype data were graciously provided by the Center for Genome Science, Korea National Institute of Health. DNA samples were isolated from the peripheral blood samples from participants and the genotype data was produced by the Korea Biobank Array (Affymetrix, Santa Clara, CA, USA) [62]. The Korea Biobank Array (referred to as Korean Chip) was optimized for the Korean population and demonstrated findings from the GWAS of blood biochemical traits. The Korean Chip comprised >833,000 markers including >247,000 rare-frequency or functional variants estimated from >2500 sequencing data in Koreans. Korean Chip achieved higher imputation performance owing to the excellent genomic coverage of 95.38% for common variants [63].

The experimental results for the Korea Biobank Array were filtered using quality control procedures based on the following criteria: call rate higher than 97%, minor allele frequency higher than 1%, and no significant deviation from the Hardy-Weinberg equilibrium (*p* < 1 × 10^−5^). After quality control assessment, the experimental genotypes were used in the imputation genotype dataset of the 1000 Genomes Phase 1 and 2 Asian panel. After following this protocol, 7,975,321 SNPs (chromosomes 1 to 22) were available for the GWAS.

### 4.5. Statistical Analysis

Baseline characteristics of the study population were presented as mean with standard deviation of the mean values for continuous variables and numbers with proportions for categorical variables. Clinical characteristics were compared between subjects with and without each metabolic disorder using t-tests and chi-square tests. Additionally, we compared the population characteristics between the patients with metabolic syndrome and the normal group and described them in Table 1.

Multivariate logistic regression analyses adjusted for sex (only for total sample GWAS), age, and lifestyle behaviors (alcohol consumption, smoking, and exercise) as covariates via additive models were used in the GWAS analysis for each metabolic disorder, implemented in PLINK version 1.9 [64]. We excluded the subjects taking medications for dyslipidemia in the analyses for types of dyslipidemia (hypercholesterolemia, hypertriglyceridemia, and hypo-HDL cholesterolemia). Manhattan and Miami plots [65] were drawn using R 4.0.2 (22 June 2020, https://cran.r-project.org/bin/windows/base/). Manhattan plots were illustrated using −log10 *p*-values on for chromosome position, and Miami plots illustrated the female and male GWAS results. Miami plots were designed at the University of Michigan to describe two comparable GWAS results [65]. Subsequently, the GWASs were performed again by subgroups according to sex and age groups. Significant associations were defined by genome-wide significance levels (*p*-values < 5 × 10^−8^). According to the plots for each metabolic disorder, we considered the groups with significant signals as the loci for each disorder. Each locus was numbered using the ‘L’ character, which indicated the significant locus. The same ‘L’ characters and numbering between Manhattan and Miami plots indicated the same locus, which implied that the locus was similarly associated with the phenotype in both the total group and subgroup. We extracted the most significant SNP (top signal) for each locus (Appendix A). For each loci with high significance SNPs through GWAS analysis, LD analysis was performed between significant SNPs, and the SNP with the most significant *p*-value among SNPs with LD r2 > 0.8 was selected as the Top Significant SNP. LD analysis was performed using Locuszoom web-based software (http://locuszoom.org/).

## Figures and Tables

**Figure 1 ijms-23-11889-f001:**
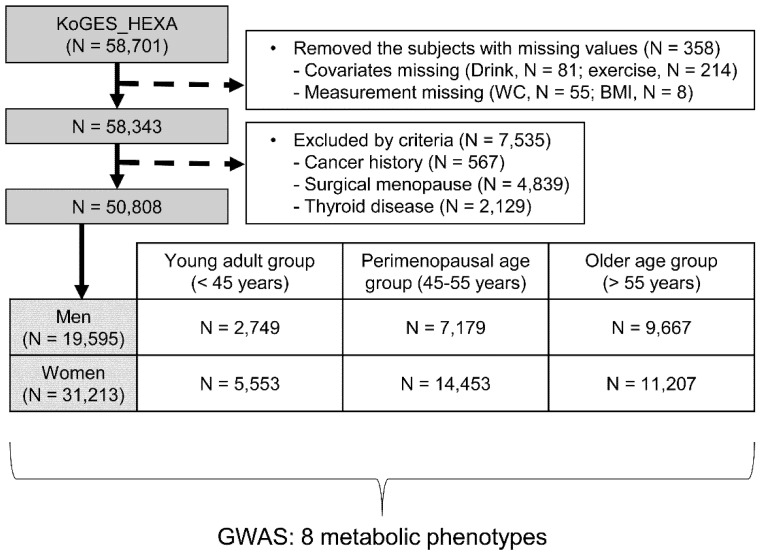
Flow chart of the study. The perimenopausal age group (45–55 years) includes perimenopausal women and perimenopause-corresponding men. KoGES, Korean Genome and Epidemiology Study; WC, waist circumference; BMI, body mass index.

**Figure 2 ijms-23-11889-f002:**
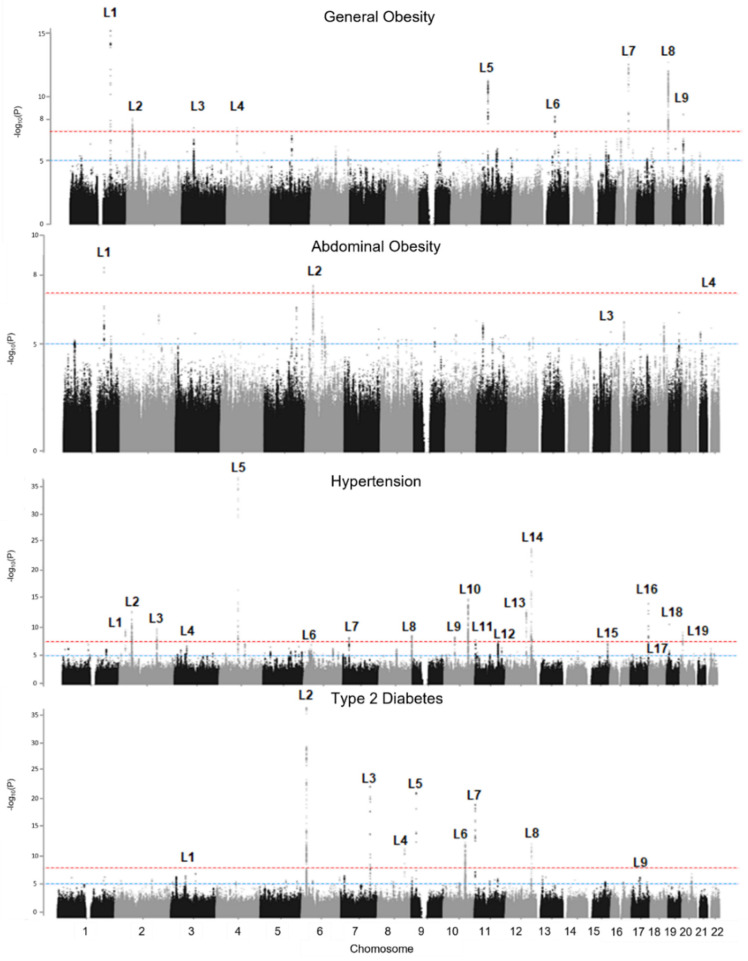
Manhattan plots of each GWAS *p*-value. The X-axis is the number of chromosomes, and the Y-axis is the minus Log_10_(P). Each locus was numbered using the ‘L’ character, which indicated the significant locus. The same ‘L’ characters and numbering between Manhattan and Miami plots indicated the same locus, which implied that the locus was similarly associated with the phenotype in both the total group and subgroup. We extracted the most significant SNP (top signal) for each locus.

**Figure 3 ijms-23-11889-f003:**
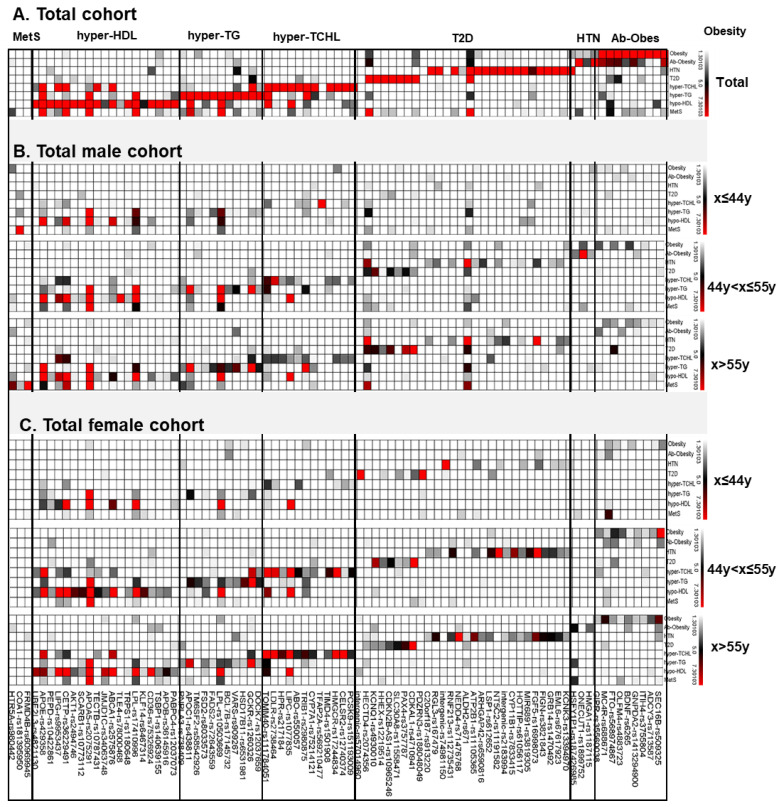
Heat-map analysis based on genome-wide association study results. A red mark means an association with a *p*-value of less than 5 × 10^−8^, and black or gray means a case that shows a *p*-value between 0.05 and 5 × 10^−8^, although not 5 × 10^−8^ significance. (**A**) shows the significant results confirmed in the GWAS analysis using the entire sample, (**B**) shows the results for the male group, and (**C**) shows the results for the female group.

**Table 1 ijms-23-11889-t001:** General characteristics of the study population.

Variables	Total	Metabolic Syndrome			Whole Subjects			Young Adult Group			Perimenopausal Age Group			Older Age Group		
		Case	Control	*p*	Men	Women	*p*	Men	Women	*p*	Men	Women	*p*	Men	Women	*p*
*N*	50,808	8115	42,693		19,595	31,213		2749	5553		7179	14,453		9667	11,207	
Age (years)	53.6 ± 8.1	55.95 ± 7.87	53.14 ± 8.09	<0.0001	55.1 ± 8.4	52.7 ± 7.8	<0.0001	41.6 ± 1.5	41.6 ± 1.5	0.375	50.6 ± 3.1	50.3 ± 3.0	<0.0001	62.2 ± 4.2	61.2 ± 3.9	<0.0001
**Clinical Measurements**																
Height (cm)	161.2 ± 8.0	162.7 ± 8.6	161.0 ± 7.9	<0.0001	168.7 ± 5.7	156.5 ± 5.2	<0.0001	171.5 ± 5.6	158.8 ± 5.0	<0.0001	169.5 ± 5.5	157.0 ± 5.0	<0.0001	167.3 ± 5.5	154.9 ± 5.1	<0.0001
Weight (kg)	62.3 ± 10.0	69.8 ± 10.6	60.9 ± 9.3	<0.0001	69.6 ± 9.0	57.7 ± 7.7	<0.0001	72.2 ± 9.9	57.4 ± 8.1	<0.0001	70.7 ± 9.0	57.7 ± 7.6	<0.0001	68.1 ± 8.6	58.0 ± 7.5	<0.0001
BMI (kg/m^2^)	23.9 ± 2.9	26.3 ± 2.9	23.4 ± 2.6	<0.0001	24.4 ± 2.7	23.6 ± 2.9	<0.0001	24.5 ± 2.9	22.8 ± 3.0	<0.0001	24.6 ± 2.7	23.4 ± 2.8	<0.0001	24.3 ± 2.6	24.2 ± 2.9	<0.001
Waist (cm)	81.0 ± 8.7	89.0 ± 7.63	79.5 ± 8.0	<0.0001	85.7 ± 7.4	78.0 ± 8.1	<0.0001	84.6 ± 8.6	75.2 ± 8.2	<0.0001	85.5 ± 8.2	77.1 ± 7.9	<0.0001	86.0 ± 7.9	80.5 ± 8.2	<0.0001
SBP (mmHg)	122.5 ± 14.8	133.7 ± 14.3	120.4 ± 13.9	<0.0001	125.7 ± 14.0	120.6 ± 14.9	<0.0001	124.0 ± 13.4	114.1 ± 13.4	<0.0001	124.4 ± 14.3	119.5 ± 14.8	<0.0001	126.9 ± 14.4	125.0 ± 14.9	<0.0001
DBP (mmHg)	75.9 ± 9.8	82.0 ± 9.6	74.7 ± 9.4	<0.0001	78.3 ± 9.5	74.3 ± 9.6	<0.0001	78.1 ± 10.0	70.9 ± 9.3	<0.0001	78.5 ± 10.1	74.0 ± 9.8	<0.0001	78.2 ± 9.4	76.2 ± 9.3	<0.0001
FPG (mg/dl)	95.3 ± 19.9	110.0 ± 29.4	92.4 ± 16.0	<0.0001	99.3 ± 22.5	92.7 ± 17.6	<0.0001	92.2 ± 24.0	87.2 ± 20.1	<0.0001	96.2 ± 28.6	89.2 ± 22.6	<0.0001	97.7 ± 29.2	93.5 ± 24.8	<0.0001
Total cholesterol (mg/dl)	197.0 ± 35.6	200.42 ± 37.79	196.4.0 ± 35.2	<0.0001	192.4 ± 34.8	199.9 ± 35.8	<0.0001	195.5 ± 34.3	186.0 ± 31.3	<0.0001	195.6 ± 34.2	200.5 ± 35.2	<0.0001	189.3 ± 35.0	205.9 ± 36.9	<0.0001
Triglyceride (mg/dl)	126.0 ± 86.7	215.6 ± 123.4	108.9 ± 65.0	<0.0001	148.0 ± 102.4	112.1 ± 71.7	<0.0001	159.7 ± 114.8	93.6 ± 65.8	<0.0001	156.1 ± 110.2	108.3 ± 69.3	<0.0001	138.5 ± 91.2	125.4 ± 75.3	<0.0001
HDL cholesterol (mg/dl)	53.5 ± 13.1	43.3 ± 9.4	55.5 ± 12.9	<0.0001	49.2 ± 11.9	56.3 ± 13.2	<0.0001	49.1 ± 11.1	58.2 ± 13.2	<0.0001	49.0 ± 11.7	57.3 ± 13.4	<0.0001	49.3 ± 12.2	54.1 ± 12.5	<0.0001
**Metabolic phenotypes**																
General obesity	16,429 (32.3%)	5389 (66.4%)	11,040 (1.3%)	<0.0001	7813 (39.9%)	8616 (27.6%)	<0.0001	1120 (40.7%)	1141 (20.5%)	<0.0001	3000 (41.8%)	3575 (24.7%)	<0.0001	3693 (38.2%)	3900 (34.8%)	<0.0001
Abdominal obesity	8115 (16.0%)	5108 (62.9%)	5446 (0.6%)	<0.0001	4083 (20.8%)	4032 (12.9%)	<0.0001	489 (17.8%)	328 (5.9%)	<0.0001	1497 (20.9%)	1426 (9.9%)	<0.0001	2097 (21.7%)	2278 (20.3%)	0.045
Hypertension	13,413 (26.4%)	4349 (53.6%)	9064 (1.1%)	<0.0001	6565 (33.5%)	6848 (21.9%)	<0.0001	466 (17.0%)	327 (5.9%)	<0.0001	1971 (27.5%)	2458 (17.0%)	<0.0001	4128 (42.7%)	4063 (36.3%)	<0.0001
Type 2 diabetes mellitus	4364 (8.6%)	1919 (23.6%)	2445 (0.3%)	<0.0001	2434 (12.4%)	1930 (6.2%)	<0.0001	121 (4.4%)	113 (2.0%)	<0.0001	761 (10.6%)	634 (4.4%)	<0.0001	1552 (16.1%)	1183 (10.6%)	<0.0001
Hypercholesterolemia	5686 (11.2%)	1148 (14.1%)	4538 (0.5%)	<0.0001	1652 (8.4%)	4034 (12.9%)	<0.0001	267 (9.7%)	292 (5.3%)	<0.0001	667 (9.3%)	1823 (12.6%)	<0.0001	718 (7.4%)	1919 (17.1%)	<0.0001
Hypertriglyceridemia	6446 (12.7%)	3530 (43.5%)	2916 (0.3%)	<0.0001	3757 (19.2%)	2689 (8.6%)	<0.0001	640 (23.3%)	272 (4.9%)	<0.0001	1595 (22.2%)	1111 (7.7%)	<0.0001	1522 (15.7%)	1306 (11.7%)	<0.0001
Hypo-HDL cholesterolemia	14,320 (28.2%)	5490 (67.7%)	8830 (1%)	<0.0001	4036 (20.6%)	10,284 (32.9%)	<0.0001	503 (18.3%)	1502 (27.0%)	<0.0001	1508 (21.0%)	4372 (30.2%)	<0.0001	2025 (20.9%)	4410 (39.4%)	<0.0001
Metabolic syndrome	10,554 (20.8%)	-	-	-	5025 (25.6%)	5529 (17.7%)	<0.0001	618 (22.5%)	614 (11.1%)	<0.0001	1788 (24.9%)	2016 (13.9%)	<0.0001	2619 (27.1%)	2899 (25.9%)	0.016
**Lifestyle habit**																
Smoker	14,976 (29.5%)	8108 (99.9%)	42,664 (5.1%)	<0.0001	14,001 (71.5%)	975 (3.1%)	0.094	2023 (73.6%)	252 (4.5%)	0.201	5241 (73.0%)	483 (3.3%)	0.079	6737 (69.7%)	240 (2.1%)	0.127
Alcohol consumer	25,729 (50.6%)	4308 (53.1%)	21,421 (2.5%)	0.569	15,533 (79.3%)	10,196 (32.7%)	<0.0001	2323 (84.5%)	2663 (48.0%)	<0.0001	5861 (81.6%)	5206 (36.0%)	<0.0001	7349 (76.0%)	2327 (20.8%)	<0.0001
Routine exercise	27,672 (54.5%)	4178 (51.5%)	23,494 (2.8%)	<0.0001	11,570 (59.0%)	16,102 (51.6%)	<0.0001	1385 (50.4%)	2363 (42.6%)	<0.0001	4208 (58.6%)	7837 (54.2%)	<0.0001	5977 (61.8%)	5902 (52.7%)	<0.0001

Perimenopausal age group (45–55 years) includes perimenopausal women and perimenopause-corresponding men. Data are expressed as mean ± SD or number (proportion). HDL, high-density lipoprotein.

**Table 2 ijms-23-11889-t002:** The significant genetic risk factors according to sex and each metabolic disorder.

	Total (Male + Female)	Total Male	Total Female
**General obesity**	*SEC16B, ADCY3, **ITIH4**, GNPDA2, BDNF, OLFM4, FTO, MC4R, GIPR*		*SEC16B, FTO, MC4R*
**Abdominal obesity**	*SEC16B, HMGA1, **XKR3***		
**Hypertension**	*KCNK3, EML6, GRB14, FIGN, FGF5, HOTTIP, CYP11B1, NT5C2, ARHGAP42, ATP2B1, ALDH2, RNF213, **LRRC30**, C20orf187*	*FGF5, ALDH2*	*EML6, GRB14, FIGN, FGF5, **HLAB**, CYP11B1, NT5C2, LSP1, ATP2B1, RNF213, **LRRC30***
**Type 2 diabetes**	*CDKAL1, PAX4, SLC30A8, CDKN2B-AS1, HHEX, KCNQ1, HECTD4*	*CDKAL1, PAX4, CDKN2B-AS1, KCNQ1, HECTD4*	*CDKAL1, PAX4, SLC30A8, CDKN2B-AS1*
**Hypercholesterolemia**	*PCSK9, CELSR2, APOB, GCKR, HMGCR, TIMD4, CYP7A1, TRIB1,ABCA1ABO, TECTB, APOA5, LIPC, CETP, HPR, LIPG, LDLR, TOMM40*	*APOB, CETP, LIPG, LDLR, TOMM40*	*PCSK9, CELSR2, APOB, GCKR, HMGCR, TIMD4, TRIB1,ABCA1ABO, APOA5, LIPC, CETP, HPR, LDLR, TOMM40,*
**Hypertriglyceridemia**	*DOCK7, GCKR*, ***HSD17B11***,***VARS**, BCL7B, LPL, TRIB1, FADS2, APOA5, **FSD2**, TM6SF2, APOC1, PNPLA3*	*DOCK7, GCKR, **VARS**, BCL7B, LPL, TRIB1, APOA5,ALDH2, APOC1*	*GCKR, **HSD17B11**, LPL, APOA5, APOC1*
**Hypo-HDL cholesterolemia**	*PABPC4, APOB, **TSBP1**, CD36, LPL, **TRPS1**, ABCA1,TECTB, APOA5, SCARB1, AKT1, LIPC, CETP, LIPG, PEPD, APOE, UBE2L3*	*LPL, ABCA1, APOA5, LIPC, CETP, LIPG, APOE*	*APOB, CD36, KLF14, LPL, ABCA1, JMJD1C, APOA5, SCARB1, AKT1, LIPC, CETP, LIPG, APOE, UBE2L3*
**Metabolic syndrome**	*LPL, APOA5, ALDH2, CETP, APOC1*	** * HTR5A * ** *, LPL, APOA5, ALDH2, CETP, APOC1*	*APOA5*

Note. The genes with bold-underline indicate that the findings not previously reported for the metabolic disorders.

**Table 3 ijms-23-11889-t003:** The significant genetic risk factors according to the age and sex subgroups.

	Young Adult Group	Peri-Menopause (Corresponding)	Older Adult Group
**Men**			
GenOb			
AbdOb			
HT			
T2D			*CDKAL1*
TC	*TFAP2A*		*APOB*
TG	*APOA5*	*GCKR, LPL, APOA5, APOC1*	*GCKR, TRIB1, APOA5, APOC1*
HDL	*APOA5*	*LPL, **intergenic (9q21.31)**, APOA5, LIPC, CETP, APOE*	*LPL, APOA5, LIPC, CETP, APOE*
MetS	** * COA1 * **	*APOA5*	** * FRMD4B * ** *, APOA5*
**Women**			
GenOb		*SEC16B*	*SEC16B, MC4R*
AbdOb		*ONECUT1*	
HT			
T2D	** * intergenic (3p13), intergenic (7p12) * **	*CDKAL1*	*CDKAL1*
TC	*APOB*	*APOB, HMGCR, LIPC, TOMM40*	*CELSR2, APOB, HMGCR, LIPC, LDLR, TOMM40*
TG	*APOA5*	*GCKR, APOA5*	*GCKR, APOA5*
HDL	*LPL, APOA5, LIPC, CETP*	*LPL, ABCA1, APOA5, LIPC, CETP, LIPG, APOE*	*CD36, LPL, ABCA1, APOA5, LIPC, CETP, LIPG*
MetS		*APOA5*	*APOA5*

Note. The genes with bold-underline indicate that the findings not previously reported for the metabolic disorders.

**Table 4 ijms-23-11889-t004:** Summary of newly identified loci among the significant results with metabolic factors based on menopausal age.

Group.Locus	Chr.BP ^a^	SNP ^b^	Gene	M ^c^	m ^d^	MAF	Total		Significant Group		
							OR	*p*	Subgroup	OR	*p*
** *General obesity* **											
go-L3	3:52866289	rs3755804	*ITIH4*	C	T	0.22	1.1	** 2.7 × 10^−8^ **			
** *Abdominal obesity* **											
ab-L4	22:17308870	rs187426985	*XKR3*	A	G	0.02	1.4	** 4.5 × 10^−8^ **			
** *Hypertension* **											
htn-L6	6:31322144	rs3819305	*HLAB*	C	G	0.42	0.9	1.8 × 10^−6^	Total Female	0.9	** 2.8 × 10^−8^ **
htn-L18	19:11526765	rs167479	*LRRC30*	G	T	0.48	0.9	** 5.4×10−11 **	Total Female	0.9	** 2.2 × 10^−8^ **
** *Type 2 diabetes mellitus* **											
dm-L1	3:73842084	rs188048049	*intergenic (3p13)*	G	A	0.01	1.0	9.8 × 10^−1^	Young adult-female	5.9	** 1.2 × 10^−8^ **
dm-L9	17:13082141	rs57014960	*intergenic (7p12)*	T	A	0.10	1.1	5.3 × 10^−2^	Young adult-female	3.8	** 1.0 × 10^−8^ **
** *Hypertriglyceridemia* **											
tg-L3	4:88284096	rs6531981	*HSD17B11*	T	A	0.32	1.1	** 8.9×10−10 **	Total Female	1.2	**1.2 × 10^−8^**
tg-L4	6:31746548	rs909267	*VARS*	T	C	0.05	1.3	** 3.3×10−16 **	Total Male	1.3	** 3.8 × 10^−12^ **
tg-L11	15:83434410	rs8033573	*FSD2*	G	A	0.50	0.9	** 8.6 × 10^−9^ **			
** *Hypo-HDL cholesterolemia* **											
hdl-L3	6:32278521	rs140639155	*TSBP1*	CA	C	0.15	0.9	** 9.2 × 10^−9^ **			
hdl-L7	8:116471025	rs1180648	*TRPS1*	G	T	0.16	1.1	** 2.3×10−9 **			
hdl-L8	9:81970758	rs78000468	*intergenic (9q21.31)*	T	C	0.01	1.1	3.3 × 10^−3^	Peri-menopausal-male	1.7	** 2.2 × 10^−8^ **
** *Metabolic syndrome* **											
mets-L1	3:69417665	rs60969945	*FRMD4B*	C	A	0.10	1.0	1.9 × 10^−1^	Older adult-male	1.4	** 4.3 × 10^−8^ **
mets-L2	7:43701891	rs181395950	*COA1*	C	T	0.01	1.0	6.2 × 10^−1^	Young adult-male	3.2	** 2.4 × 10^−8^ **
mets-L3	7:154876342	rs980442	*HTR5A*	G	T	0.16	0.9	9.5 × 10^−5^	Total Male	0.8	** 2.1 × 10^−8^ **

Note. Odds ratios indicate the magnitude of the influence of having a minor allele of a genetic mutation. (a) Chromosome and position based on human genome version hg19; (b) top significant SNP of the mapped loci; (c) Major allele; (d) Minor allele. Bold-underline *p*-value indicated the genome-wide significance level (*p*-value < 5 × 10^-8^) and the boxed *p*-value indicated that they passed the bonferroni correction *p*-value (*p*-value < 6.27 × 10^-9^).

## Data Availability

The data are not publicly available due to the policy of KCDC.

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
