# Peer review of "Lifestage Sex-Specific Genetic Effects on Metabolic Disorders in an Adult Population in Korea: The Korean Genome and Epidemiology Study"

_ijms, 2022, doi:10.3390/ijms231911889_

Round 1

Reviewer 1 Report

Congratulation! Great work!

The genome-wide association studies (GWASs) was evaluated for the association of genetic markers (genetic risk factors) to metabolic phenotypes such as: general obesity, abdominal obesity, hypertension, type 2 diabetes, hypercholesterolemia, hypertriglyceridemia, hypo-high-density lipoprotein cholesterolemia, and metabolic syndrom for a specific population (Korean people) -  17 loci (: GO-L3:ITIH4-MUSTN1 for general obesity;AB-L4:XKR3 for abdominal obesity; HTN-L6:HLAB and HTN-L18:LRRC30 for HTN; DM-L1:3p13 and DM-L9:7p12 for T2D; TG-L3:HSD17B13-NUDT9, TG-L4:VWA7-VARS, and TG-L11:SNHG21-FSD2 for hypertriglyceridemia; HDL-L3:TSBP1, HDL-L7:TRPS1, and HDL-L8:9q21.31 for hypo- HDL cholesterolemia; MetS-L1:6q21, MetS-L2:COA1, and MetS-L3:HTR5A for MetS.)  were the first report to be associated with risk of metabolic disorder in Korean population.

Observation:

Figure 2 B and 2 C have to be with same size like 2A (or have to be more clear / more pixels for figures 2B and 2C) in order to be clear the text from right and from down (even if it is similar with 2A).

Author Response

Response to Reviewer 1’s Comments

Manuscript ID: ijms-1913759

Title: Lifestage Sex-Specific Genetic Effects on Metabolic Disorders in an Adult Population in Korea: The Korean Genome and Epidemiology Study

Authors: Young-Sang Kim, Yon Chul Park, Ja-Eun Choi, Jae-Min Park, Kunhee Han, Kwangyoon Kim, Bom-Taeck Kim, Kyung-Won Hong

The genome-wide association studies (GWASs) was evaluated for the association of genetic markers (genetic risk factors) to metabolic phenotypes such as: general obesity, abdominal obesity, hypertension, type 2 diabetes, hypercholesterolemia, hypertriglyceridemia, hypo-high-density lipoprotein cholesterolemia, and metabolic syndrom for a specific population (Korean people) -  17 loci (: GO-L3:ITIH4-MUSTN1 for general obesity;AB-L4:XKR3 for abdominal obesity; HTN-L6:HLAB and HTN-L18:LRRC30 for HTN; DM-L1:3p13 and DM-L9:7p12 for T2D; TG-L3:HSD17B13-NUDT9, TG-L4:VWA7-VARS, and TG-L11:SNHG21-FSD2 for hypertriglyceridemia; HDL-L3:TSBP1, HDL-L7:TRPS1, and HDL-L8:9q21.31 for hypo- HDL cholesterolemia; MetS-L1:6q21, MetS-L2:COA1, and MetS-L3:HTR5A for MetS.)  were the first report to be associated with risk of metabolic disorder in Korean population.

Response to Reviewer 1: Thank you very much for your kind comments on our manuscript.

Observation:

Minor Comment 1: Figure 2 B and 2 C have to be with same size like 2A (or have to be more clear / more pixels for figures 2B and 2C) in order to be clear the text from right and from down (even if it is similar with 2A).

Response to Minor Comment 1: [Figure 3] Thanks for the comment. According to the reviewer's opinion, the sizes B and C in Figure 2 were increased and changed.

Reviewer 2 Report

The title: "Lifestage Sex-Specific Genetic Effects on Metabolic Disorders in an Adult Population in Korea: The Korean Genome and Epidemiology Study" manuscript are well designed and manipulated by grouping of important factors age and sex. The results showed that 101 significant loci were associated with eight metabolic phenotypes, and 17 loci among them, will be the first report. 

Major concern

1. General obesity was defined as BMI ≥ 25 kg/m2, but there is no defined for control subjects (for example: 18.5 <= BMI <25). Therefore, Is there any possibility that underweight participants were included in the control group?

2. The 17 newly identified SNPs are proposed to be inserted into the main text as individual table.

Minor concern

1. Write the name of the gene in the text in italics.

2. 2.3 Study group-specific associations section below

   the older adultmfemale group -> the older adult female group ?

Author Response

Response to Reviewer 2’s Comments

Manuscript ID: ijms-1913759

Title: Lifestage Sex-Specific Genetic Effects on Metabolic Disorders in an Adult Population in Korea: The Korean Genome and Epidemiology Study

Authors: Young-Sang Kim, Yon Chul Park, Ja-Eun Choi, Jae-Min Park, Kunhee Han, Kwangyoon Kim, Bom-Taeck Kim, Kyung-Won Hong

The title: "Lifestage Sex-Specific Genetic Effects on Metabolic Disorders in an Adult Population in Korea: The Korean Genome and Epidemiology Study" manuscript are well designed and manipulated by grouping of important factors age and sex. The results showed that 101 significant loci were associated with eight metabolic phenotypes, and 17 loci among them, will be the first report. 

 Response to Reviewer 2 : Thank you very much for your kind comments on our manuscript.

Major concern

Major Comment 1: General obesity was defined as BMI ≥ 25 kg/m2, but there is no defined for control subjects (for example: 18.5 <= BMI <25). Therefore, Is there any possibility that underweight participants were included in the control group?

Response : %of BMI < 18.5

Response to Major Comment 1:  Thank you for your comments on the important point. Although it is true that the risk of death is high at underweight with a BMI of less than 18.5 (Ref 1), the population in this cohort is less than 2%. Because underweight does not mean that it is metabolically unhealthy, it is thought that there is no difficulty in expressing the genetic characteristics of obesity as reported in this study. Also, although some literature tends to define a higher BMI as obesity (Ref 2), the Korean Society for Obesity still considers BMI 25 as the cutoff for obesity (Ref 3).

Ref 1. Jee, Sun Ha, et al. "Body-mass index and mortality in Korean men and women." New England Journal of Medicine 355.8 (2006): 779-787.

Ref 2. Jih, Jane, et al. "Using appropriate body mass index cut points for overweight and obesity among Asian Americans." Preventive medicine 65 (2014): 1-6.

Ref 3. Kim, Bo-Yeon, et al. "2020 Korean Society for the Study of Obesity guidelines for the management of obesity in Korea." Journal of obesity & metabolic syndrome 30.2 (2021): 81-92.

Major Comment 2: The 17 newly identified SNPs are proposed to be inserted into the main text as individual table.

 Response to Major Comment 2: Thank you for your suggestion. We added the individual table for the newly identified SNPs in the Table 4. In the Original Manuscript, we tried to express all significant results in one Table 2, but we thought that the table was too large and there were many difficulties to understand. Therefore, the Tables were divided into three. In Table 2 and Table 3, the genes with statistical significance for each group were summarized and expressed, and the newly discovered genes are separately indicated in Table 4. The original Table 2 is moved to Supplementary Table 1. One thing to apologize is that we previously stated that the number of newly discovered genes was 17, but when we revise the manuscript, it was 15. We corrected all the number of newly discovered genes throughout the paper.

Minor concern

Minor Comment 1: Write the name of the gene in the text in italics.

 Response to Minor Comment 1: Thanks for the important comment. Gene names have been corrected in italics throughout the paper.

Minor Comment 2: 2.3 Study group-specific associations section below the older adultmfemale group -> the older adult female group ?

 Response to Minor Comment 2: Thanks for the important comment. As you said, the word was Typo and it was corrected to the older adult female group

Reviewer 3 Report

Dear Authors,

Thank you for the opportunity to review this paper. I am experienced in this field working on several papers of this kind. In relevance to this paper, my main expertise is in GWAS analytics and genomic data mining.

The authors present a very interesting approach to evaluating a large population cohort in Korea. The study is composed of sub stratified analysis by sex and age groups that can be advantageous to detect targets that are relevant to each of them. Overall, the paper is written with a clear language. However, there are many things that are not well described. It gives me the impression that several sections were written by different people without them looking at the whole for flow and context.

MAIN CONCERNS

Methods

Obesity at BMI ≥25? Is this the Korean standard? Some references should be included as this is not the standard in other parts of the world.

It is unclear of what the data is, what are the characteristics of the array or arrays used. They mention another paper, but that information on the details of the platform(s) used and quality control parameters is crucial and should be in the paper. Were CNV regions excluded or not? Were low frequency variants excluded? What about populations structure correction?

Imputation was used to achieve a 7,975,321 SNP dataset for the GWAS, it is unclear of how much was data from the arrays and how much was imputed. By the way, I would like to note that the 5x10-8 threshold is less stringent than the 6.27x10-9 Bonferroni threshold. Therefore, the significance threshold is in my opinion on the relaxed side.

The baseline characteristics appear to have been evaluated between subjects with and without metabolic disorder syndrome using t tests and chi square tests, where is that reported in the paper? This is one of those comments that make me thing of that multiple writer disconnection.

The GWAS methodology is unclear and it is uncertain of how things were actually analyzed. The paper mentions that a multivariate logistic regressions adjusted for sex, age and lifestyle was used; how was this done if the main strategy of the study is to stratify by sex and age? I have no idea of exactly what is that lifestyle variable.

LD investigation was mentioned, but is not formally presented. Again, something disconnected of the narrative.

There is no second cohort, this is noted in the discussion but it is something that could have been implemented easily considering the size of the sample. Even using a 10% subset cohort for this purpose would have been better.

Results

There are one figure and one table in the main paper with a series of Manhattan plots and Miami plots as supplementary. The supplementary looks as a figure dump that is presented without any discussion or context. Some panels is the figures are so small that they are unreadable. Some of these Manhattan plots should be in the main paper along with some fine mapping to highlight the main peaks. Here is where the LD investigation mentioned would shine. The fine mapping and LD block evaluation can provide strong evidence of the validity of the SNPs. 

The table in the main text is a compilation of descriptive statistics. These are discussed in the text. It is unclear if the arguments mentioned in the text are supported by formal tests. The table should include significance testing.

The figure displays a type of summary heatmap that has all the results from the study compiled. The scale is unclear, what are the units? ORs? And the shading is confusing white to green to red. I have a slightly altered color vision, these colors here are not hard for me but I know several colleagues with more severe color vision impairments who would be completely unable to distinguish anything in the figures. Panel B and C are unreadable .

There is no summary list for the SNPs detected, these should have chromosome number, alleles, MAF, P-values, gene annotations. This table is found in pretty much all GWAS papers and is a major draw of attention to all readers interested.

Discussion

The discussion is very disorganized, some subheadings on the arguments presented would be very useful. For example: talk about sex related findings, then age findings then combined. 

A main limitation not mentioned is the uniqueness of the Korean population. The Korean population and environment may not be representative or even relevant to other regions of the world. Discussing how the results of this study may be influenced by environmental factors would be important.

Summary and decision

The study has merit but it is definitely unacceptable in its current form. The paper is clear in its language and main message but it is very disorganized and incomplete in many areas. This paper would require a substantial major revision. 

MINOR

The methods section between the results and discussion. Never seen that before. Either place them after the intro or at the end.

Again thanks for the invitation to review this work and good luck with your research.

Author Response

Response to Reviewer 3’s Comments

Manuscript ID: ijms-1913759

Title: Lifestage Sex-Specific Genetic Effects on Metabolic Disorders in an Adult Population in Korea: The Korean Genome and Epidemiology Study

Authors: Young-Sang Kim, Yon Chul Park, Ja-Eun Choi, Jae-Min Park, Kunhee Han, Kwangyoon Kim, Bom-Taeck Kim, Kyung-Won Hong

Reviewer’s Comments

Thank you for the opportunity to review this paper. I am experienced in this field working on several papers of this kind. In relevance to this paper, my main expertise is in GWAS analytics and genomic data mining.

The authors present a very interesting approach to evaluating a large population cohort in Korea. The study is composed of sub stratified analysis by sex and age groups that can be advantageous to detect targets that are relevant to each of them. Overall, the paper is written with a clear language.

Major Comment: However, there are many things that are not well described. It gives me the impression that several sections were written by different people without them looking at the whole for flow and context.

Response to Reviewer 1: We really appreciated to your invaluable comments for our manuscript. We corrected the manuscript line by line following your detailed comments.

MAIN CONCERNS

Methods

Major Comment 1: Obesity at BMI ≥25? Is this the Korean standard? Some references should be included as this is not the standard in other parts of the world.

Response to Major Comment 1:  Thank you for your comments on the important point. Although it is true that the risk of death is high at underweight with a BMI of less than 18.5 (Ref 1), the population in this cohort is less than 2%. Because underweight does not mean that it is metabolically unhealthy, it is thought that there is no difficulty in expressing the genetic characteristics of obesity as reported in this study. Also, although some literature tends to define a higher BMI as obesity (Ref 2), the Korean Society for Obesity still considers BMI 25 as the cutoff for obesity (Ref 3).

Ref 1. Jee, Sun Ha, et al. "Body-mass index and mortality in Korean men and women." New England Journal of Medicine 355.8 (2006): 779-787.

Ref 2. Jih, Jane, et al. "Using appropriate body mass index cut points for overweight and obesity among Asian Americans." Preventive medicine 65 (2014): 1-6.

Ref 3. Kim, Bo-Yeon, et al. "2020 Korean Society for the Study of Obesity guidelines for the management of obesity in Korea." Journal of obesity & metabolic syndrome 30.2 (2021): 81-92.

Major Comment 2: It is unclear of what the data is, what are the characteristics of the array or arrays used. They mention another paper, but that information on the details of the platform(s) used and quality control parameters is crucial and should be in the paper. Were CNV regions excluded or not? Were low frequency variants excluded? What about populations structure correction?

Response to Major Comment 2: We included the characteristics of Korean Biobank chip in the Method section. “The Korea Biobank Array (referred to as Korean Chip), optimized for the Korean popula-tion and demonstrate findings from GWAS of blood biochemical traits. Korean Chip comprised >833,000 markers including >247,000 rare-frequency or functional variants estimated from >2,500 sequencing data in Koreans. Korean Chip achieved higher im-putation performance owing to the excellent genomic coverage of 95.38% for com-mon[27].”

Major Comment 3: Imputation was used to achieve a 7,975,321 SNP dataset for the GWAS, it is unclear of how much was data from the arrays and how much was imputed. By the way, I would like to note that the 5x10-8 threshold is less stringent than the 6.27x10-9 Bonferroni threshold. Therefore, the significance threshold is in my opinion on the relaxed side.

Response to Major Comment 3: We described the significant association at two different criteria. One criterion is the general GWAS significance criteria (5x10-8) and the other criterion is the Bonferroni Threshold significant criteria (6.27 x 10-9).

Table 4. GWAS significance is bold and underlined, Bonferroni criteria is bold and boxed.

Page XX. Lin XX separated the significance to two different criteria.

Major Comment 4: The baseline characteristics appear to have been evaluated between subjects with and without metabolic disorder syndrome using t tests and chi square tests, where is that reported in the paper? This is one of those comments that make me thing of that multiple writer disconnection.

Response to Major Comment 4: As suggested by the reviewer, we performed a comparative analysis between patients with metabolic syndrome and normal people, the method was described in the Method section, and the results are included in Table 1.

Major Comment 5: The GWAS methodology is unclear and it is uncertain of how things were actually analyzed. The paper mentions that a multivariate logistic regressions adjusted for sex, age and lifestyle was used; how was this done if the main strategy of the study is to stratify by sex and age? I have no idea of exactly what is that lifestyle variable.

Response to Major Comment 5: We corrected the method section. “Multivariate logistic regression analyses adjusted for sex(only for total sample GWAS), age, and lifestyle behaviors(Alcohol Consumption, Smoking, and Exercise) as covariates via additive models were used in the GWAS analysis for each metabolic dis-order, implemented in PLINK version 1.9 [28].”

Major Comment 6: LD investigation was mentioned, but is not formally presented. Again, something disconnected of the narrative.

Response to Major Comment 6: As the reviewer’s suggestion, we corrected the LD investigation methods. “For each loci with high significance SNPs through GWAS analysis, LD analysis was performed between significant SNPs, and the SNP with the most significant p-value among SNPs with LD r2 > 0.8 was selected as the Top Significant SNP. LD analysis was performed using Locuszoom web-based software (http://locuszoom.org/)”. Additionally, we created the signal plot for the fine mapping and displayed in a new supplementary Figure 9).

Major Comment 7: There is no second cohort, this is noted in the discussion but it is something that could have been implemented easily considering the size of the sample. Even using a 10% subset cohort for this purpose would have been better.

Response to Major Comment 7: Thanks for the very nice comment. We positively agree with the reviewer's opinion, but if the samples are grouped like our research model, the number of samples for each group is too reduced, so the 10% sampling test is difficult in this study. We will apply the method in a larger study group.

Results

Major Comment 8: There are one figure and one table in the main paper with a series of Manhattan plots and Miami plots as supplementary. The supplementary looks as a figure dump that is presented without any discussion or context. Some panels is the figures are so small that they are unreadable. Some of these Manhattan plots should be in the main paper along with some fine mapping to highlight the main peaks. Here is where the LD investigation mentioned would shine. The fine mapping and LD block evaluation can provide strong evidence of the validity of the SNPs. 

Response to Major Comment 8: As the reviewer’s suggestion, we corrected the LD investigation methods. “For each loci with high significance SNPs through GWAS analysis, LD analysis was performed between significant SNPs, and the SNP with the most significant p-value among SNPs with LD r2 > 0.8 was selected as the Top Significant SNP. LD analysis was performed using Locuszoom web-based software (http://locuszoom.org/)”. Additionally, we created the signal plot for the fine mapping and displayed in a new supplementary Figure 9).

Major Comment 9: The table in the main text is a compilation of descriptive statistics. These are discussed in the text. It is unclear if the arguments mentioned in the text are supported by formal tests. The table should include significance testing.

Response to Major Comment 9: We added the test statistics for baseline characteristics as your suggestion in Table 1.

Major Comment 10: The figure displays a type of summary heatmap that has all the results from the study compiled. The scale is unclear, what are the units? ORs? And the shading is confusing white to green to red. I have a slightly altered color vision, these colors here are not hard for me but I know several colleagues with more severe color vision impairments who would be completely unable to distinguish anything in the figures. Panel B and C are unreadable .

Response to Major Comment 10: [Figure 3] Thanks for the comment. According to the reviewer's opinion, the sizes B and C in Figure 2 were increased and changed the figure number to Figure 3.

Major Comment 11: There is no summary list for the SNPs detected, these should have chromosome number, alleles, MAF, P-values, gene annotations. This table is found in pretty much all GWAS papers and is a major draw of attention to all readers interested.

Response to Major Comment 11: Thanks for the comment. According to the reviewer's opinion, We added the description at Table 4.

Discussion

Major Comment 12: The discussion is very disorganized, some subheadings on the arguments presented would be very useful. For example: talk about sex related findings, then age findings then combined. 

Response to Major Comment 12: Thanks for the comment. According to the reviewer's opinion, sub heading added to the discussion section.

“4.1. Significant associations in peri- and postmenopausal women

4.2 Newly identified metabolic syndrome gene loci

4.3 Confirmed the Previous Associations”

Major Comment 13: A main limitation not mentioned is the uniqueness of the Korean population. The Korean population and environment may not be representative or even relevant to other regions of the world. Discussing how the results of this study may be influenced by environmental factors would be important.

Response to Major Comment 13: Thanks for the comment. According to the reviewer's opinion, We included it in the limitation section. “The subjects of our study is Koreans, and, therefore, the results of this study may be different in different ethnic groups due to different environmental factors. Therefore, it is necessary that the results of this study confirmed in different ethnic groups.”

Summary and decision

The study has merit but it is definitely unacceptable in its current form. The paper is clear in its language and main message but it is very disorganized and incomplete in many areas. This paper would require a substantial major revision. 

MINOR

Minor Comment 1: The methods section between the results and discussion. Never seen that before. Either place them after the intro or at the end.

 Response to Minor Comment 1: Thanks for the comment. According to the reviewer's opinion, we moved the Method section between Introduction and Results.

Again thanks for the invitation to review this work and good luck with your research.

Round 2

Reviewer 3 Report

I must admit that I am impressed with the level of improvement made by the authors in a single revision on such a short time. 

The new figures added and specially the supplementary table with the hit list elevate the paper to a high standard. Although the addition of these extra material introduced some awkward blank spaces, this can be fixed during proofing.

The authors did an outstanding job, my concerns have been satisfied and I am happy to recommend this paper for publication.

Best wished on all your future research.